# Incentive Contract Design for the Water-Rail-Road Intermodal Transportation with Travel Time Uncertainty: A Stackelberg Game Approach

**DOI:** 10.3390/e21020161

**Published:** 2019-02-09

**Authors:** Wenying Zhang, Xifu Wang, Kai Yang

**Affiliations:** 1School of Traffic and Transportation, Beijing Jiaotong University, Beijing 100044, China; zhangwenying@chinabata.cn; 2School of Mathematical Sciences, Monash University, Clayton, VIC 3800, Australia

**Keywords:** intermodal transportation, asymmetric information, stackelberg game, uncertain theory, entropy

## Abstract

In the management of intermodal transportation, incentive contract design problem has significant impacts on the benefit of a multimodal transport operator (MTO). In this paper, we analyze a typical water-rail-road (WRR) intermodal transportation that is composed of three serial transportation stages: water, rail and road. In particular, the entire transportation process is planned, organized, and funded by an MTO that outsources the transportation task at each stage to independent carriers (subcontracts). Due to the variability of transportation conditions, the travel time of each transportation stage depending on the respective carrier’s effort level is unknown (asymmetric information) and characterized as an uncertain variable via the experts’ estimations. Considering the decentralized decision-making process, we interpret the incentive contract design problem for the WRR intermodal transportation as a Stackelberg game in which the risk-neutral MTO serves as the leader and the risk-averse carriers serve as the followers. Within the framework of uncertainty theory, we formulate an uncertain bi-level programming model for the incentive contract design problem under expectation and entropy decision criteria. Subsequently, we provide the analytical results of the proposed model and analyze the optimal time-based incentive contracts by developing a hybrid solution method which combines a decomposition approach and an iterative algorithm. Finally, we give a simulation example to investigate the impact of asymmetric information on the optimal time-based incentive contracts and to identify the value of information for WRR intermodal transportation.

## 1. Introduction

Water-rail-road (WRR) intermodal transportation is defined as a system that transfers the cargo from an origin to a destination in one and the same intermodal transportation unit (e.g., a twenty-foot equivalent unit (TEU) container) without handling of the goods themselves when changing modes by using three different means of transport: water, rail and road. Compared to the unimodal one, the basic features of the WRR intermodal transportation are: (1) the entire transportation process can be divided into a series of sequential transportation stages; (2) three transportation modes are used for the carriage of cargos; (3) one party, usually called multimodal transport operator (MTO), is responsible for the entire carriage and acts as a principal. In particular, the MTO does not own any means of transport, and needs to outsource to specialized subcontractors who act as agents for all modes of transport in order to fulfill the whole intermodal transportation chain. These subcontractors (carriers) might be a waterway operator, a railway operator and a road haulier. Motivated by this, the study presented in this paper focuses on designing contracts for the carriers engaging in the WRR intermodal transportation from the MTO’s perspective.

By being an experienced organizer in making optimal combinations of different modes of transport, the MTO is responsible for the cargo-owner in the business of the WRR intermodal transportation, and for the cargo’s complete journey from its original place of dispatch to its ultimate destination. Due to the competitive environment in the market, the MTO pays more attention to reducing the door-to-door delivery time, which is the total travel time of three serial transportation stages. One of the major reasons for the fastest possible delivery is that the MTO wants the container to come back as soon as possible after the delivery of cargo so that he can have a higher utilization ratio of the container and eventually earn more freight. With such a concern, this paper aims at designing the time-based contracts offered by MTO for the carriers to satisfy the cargo-owner’s needs by providing the optimized door-to-door services with shorter delivery time.

In general, the travel time of each transportation stage depends on the respective carrier’s effort level, which is a broad concept and has been widely used in the economics and management field. In the intermodal transportation context, the effort level can be interpreted as the amount of human, material and financial resources required to effectively improve transport capacity and efficiently complete transportation task. For example, the carrier will incur a cost for assigning more drivers and upgrading means of transport to shorten the travel time. As a matter of fact, there exists information asymmetry between the MTO and the carriers because of maximizing their respective utilities in the decentralized decision-making system. That is, the effort level is not only the carrier’s private information and unknown to the MTO, but it is also at a cost to the carrier, and the carrier may not provide the effort level that the MTO desires. This may lead to conflict between the carriers and the MTO and delay the travel time. In the real situation, the carrier may ascribe the delay in delivery to the natural causes or force majeure rather than his possible “lazy” behavior because the MTO cannot directly observe his action (i.e., no sharing of information). Technically, one way of addressing this issue is for the MTO to introduce communication and information technology for cargo tracking. On the other hand, from the operational management point of view, the MTO can offer incentive contracts to the carriers based on the travel time of their transportation stages to ensure that the carriers invest adequate efforts. Moreover, designing the time-based incentive contract for WRR intermodal transportation is a decentralized decision-making process that can be analyzed as a Stackelberg game in which the risk-neutral MTO serves as the leader and the risk-averse carriers serve as the followers. With such a concern, the major purpose of this paper is to develop a Stackelberg game approach to formulate the time-based incentive contract design problem for WRR intermodal transportation.

Mathematically, the process of designing a time-based incentive contract can be modeled as a bi-level programming problem. In the real world, due to the variability of transportation conditions (e.g., accidents, failures and weather changes), the travel time of each transportation mode (stage) is not precisely known in advance, thus, probability theory, fuzzy theory and uncertainty theory have been introduced to overcome this issue. In some cases, sufficient historical data are known for travel time parameters and can be used to characterize stochastic (objective) uncertainty by using stochastic programming techniques. However, the travel times cannot be exactly predicted in advance when the MTO contracts with the carriers for the first time. That is, no information about their probability distributions is known except for the experiences and judgments of field experts, one must rely on fuzzy theory or uncertainty theory to handle the uncertain information. However, some surveys showed that subjective uncertainty cannot be modeled by fuzzy variables (numbers). This means that some real problems cannot be processed by fuzzy theory, because the possibility measure has no self-duality property. Doing away with the self-duality property in mathematics may lead to counterintuitive results for modeling uncertain travel times (see Appendix A for details). In order to deal with this type of uncertainty, Liu [1] founded an uncertainty theory that is a branch of mathematics based on normality, monotonicity, self-duality, and countable subadditivity axioms. Within the framework of uncertainty theory, we introduce a new version of utility function named mean-entropy form as a measure of risk in which the entropy is more general than variance because entropy is free from reliance on symmetrical uncertain distributions. Therefore, this paper aims to formulate an uncertain bi-level programming model for the time-based incentive contract design problem in a WRR intermodal transportation setting under expectation and entropy decision criteria.

### 1.1. Literature Review

This section presents a review of literature on the intermodal transportation problem, the contract theory and the uncertain theory.

The first stream studies the intermodal transportation problem. The intermodal transportation is defined as the transportation of a load from its origin to its destination by a sequence of at least two transportation modes, in which the transfer from one mode to the next is performed at an intermodal terminal [2]. Caris et al. [3] proposed new research themes concerning decision support for private stakeholders as well as for public actors such as policy makers and port authorities in intermodal transport. Lam and Gu [4] focused on developing innovative approaches in the area of enhanced intermodal network design provided by freight integrators, which are to address cost minimisation, transit time minimisation, and carbon footprint to better meet market needs. Verma and Verter [5] presented an analytical framework for planning rail-truck intermodal transportation of hazmats, where a bi-objective optimization model to plan and manage intermodal shipments is developed. Wang and Meng [6] considered a discrete intermodal network design problem for freight transportation, in which the network planner needs to minimize the total operating cost of carriers and hub operators under a general route choice model of intermodal operators. Abbassi et al. [7] developed a new mathematical formulation and two solution approaches for an intermodal transportation problem of agricultural products from Morocco to Europe to minimise the transportation cost and the maximal overtime to delivery products. Wang et al. [8] presented a bi-objective optimization formulation for the hub-and-spoke based road-rail intermodal transportation network design problem by taking into account the expected value criterion and the critical value criterion. For a detailed review of the intermodal transportation problem and its variations, interested readers can refer to Arias and Fuentes [9] and Crainic et al. [10].

The second stream of literature related to our work is on the contract theory. Contract theory (Nobel Memorial Prize in Economic Sciences 2016) has been highly successful and there are active research areas in economics, finance and management. For example, Hart and Moore [11] explored whether the parties can make up for this incompleteness to some extent by building into their contract a mechanism for revising the terms of trade as each party receives information about benefits and costs. Holmström [12] studied efficient contractual agreements in a principal-agent relationship under various assumptions about what can be observed, and hence contracted upon, by both parties. Bolton and Scharfstein [13] analyzed the optimal financial contract to balance the benefit of deterring predation by relaxing financial constraints against the cost of exacerbating incentive problems. Christensen et al. [14] reviewed theoretical and empirical work on financial contracting that is relevant to accounting researchers and discussed how the use of accounting information in contracts enhances contracting efficiency. Grinblatt and Titman [15] presented conditions for contract parameters that provide proper risk incentives for classes of investment strategies. Chao et al. [16] discussed two contractual agreements by which product recall costs can be shared between a manufacturer and a supplier to induce quality improvement effort. For learning contract theory, we refer the interested reader to Bolton and Dewatripont [17].

The third stream considers the uncertain theory. After the word “randomness” was used to represent probabilistic phenomena, Knight [18] and Keynes [19] started to use the word “uncertainty” to represent any non-probabilistic phenomena. The academic community also calls it Knightian uncertainty, Keynesian uncertainty, or true uncertainty. As the scientific terminology evolves, the concept of uncertainty had been interpreted either in terms of degrees of conviction [20], or as relative frequencies [21]. The latest development was uncertainty theory founded by Liu [1] and refined by Liu [22] based on normality, duality, subadditivity and product axioms. Since then, many experts and scholars began to pay attention to it and investigated it [23,24,25,26,27,28,29,30,31]. Based on uncertain theory, Liu [22] presented the uncertain programming. With the pioneering work of Liu [22], some work has been done on the extension of uncertain programming, such as in multilevel programming [32], multi-objective programming [33] and goal programming [34]. Recently, uncertain theory has been used to develop the contract theory by several authors. For example, Mu et al. [35] established an uncertain contract model for the rural migrant worker’s employment problem to maximize the enterprise’s expected utility. Wang et al. [36] presented an uncertain contract model for the price discrimination problem in labor market to maximize the employer’s expected welfare. Wu et al. [37] discussed an uncertain contract problem with multi-dimensional incomplete information based on the critical value criterion. Wang et al. [38] presented four classes of uncertain contract models in a project management setting based on the expected value and the critical value criteria. Yang et al. [39] built two classes of uncertain contract models for new product development and derived their respective optimal incentive contracts. Fu et al. [40] considered an uncertain contract problem to implement an R&D project through a menu of incentive contracts.

To clarify the novel features of this study, we detail the related research gaps of our paper as follows:This paper is distinct from the aforementioned work in that we consider an intermodal transportation decentralized setting that can be analyzed as a Stackelberg game in which the risk-neutral MTO serves as the leader and the risk-averse carriers serve as the followers from the point of view of microeconomics.This is the first work in which the WRR intermodal transportation has been considered to evaluate the uncertain travel time by domain expert’s knowledge, experience and professional feelings rather than the probability which is on the basis of large sample size.There are no works addressing the uncertain contract problem in a WRR intermodal transportation context. Hence, to fill this gap, this paper presents an uncertain bi-level programming approach to formulate the time-based incentive contract design model for WRR intermodal transportation uncertain travel time uncertainty.

### 1.2. Research Contributions and Outline

According to above-mentioned reasons, this study highlights some new points for the first time in the WRR intermodal transportation area which can be useful for both academics and practitioners. The main contributions of this study are outlined as follows:This paper employs uncertain variables to capture the characteristics of subjective uncertainty within a WRR intermodal transportation field based on uncertainty theory. By doing so, the uncertain travel time can be directly handled in the proposed modeling framework. Furthermore, the uncertain distributions are flexible and diversified forms to quantify the travel time uncertainty when no samples (sufficient historical data) are available.This paper adopts the expected utility and the mean-entropy utility functions to characterize MTO and carriers’ profits, respectively. In particular, the MTO’s objective based on the expected utility function is to maximize the expected profit. For the carriers, the entropy is used as a synonym for risk in the sense that uncertainty causes loss. Based on this, the mean-entropy utility allows the carrier to maximize the linear combination of expected value and entropy of the carrier’s profit. Under these two decision criteria, this paper presents an uncertain bi-level programming model to design the time-based incentive contracts offered by the MTO.This paper develops a hybrid solution method by combining a decomposition method and an iterative algorithm to obtain the parameters of the optimal time-based incentive contract and to further identify the value of information. More specifically, the former approach divides the original model into three sub-models by taking advantage of the structural characteristics. The latter approach is to interactively solve the upper-level and lower-level programming problems in the equivalent deterministic sub-models for deriving the analytical results of the proposed model.

The rest of the paper is organized as follows. Section 2 describes a modeling framework and presents an uncertain bi-level programming model to design the incentive contract for the WRR intermodal transportation. Section 3 derives the optimal time-based incentive contracts under symmetric and asymmetric information and explores the effect of information asymmetry. Section 4 conducts a simulation example to complement our analytical results. Finally, Section 5 provides conclusions and suggestions for future studies. Preliminaries on uncertainty theory and proofs of all theorems are relegated to the appendix for clarity of presentation.

## 2. Model Development and Description

In this section, we consider an MTO who is responsible for the entire WRR intermodal transportation consisting of three stages that can be transported by the water, rail and road carriers in sequence (see Figure 1). Aligning the conflicting goals of the MTO and carriers in the decentralized decision-making process is one of the major challenges and the purpose of most contracts. In reality, there is asymmetry information between the MTO and three carriers. That is, the carriers know their effort levels to accelerate their respective transportation stages, but the MTO cannot directly and fully observe how much efforts the carriers spend to shorten the travel time of their transportation stages. From the perspective of the carrier, the effort level can be measured by the amount of human, material and financial resources spent in improving transport capacity and accomplishing transportation task. As a result, the MTO should take the lead to design and offer incentive contracts to the carriers. In this way, we can model the contracting process as a Stackelberg game. Therefore, the sequence of the events in our model is described as follows:In Step 1, the MTO offers a take-it-or-leave-it incentive contract to three carriers simultaneously.In Step 2, three carriers accept or reject their contracts. If they accept, the water carrier first chooses his effort level. Then, the rail carrier chooses his effort level after the water carrier has completed his transportation stage, and the road carrier chooses his effort level after the rail carrier has completed his transportation stage. If they reject, go back to Step 1.In Step 3, the MTO pays the carriers based on their realized travel times.

Next, we propose an uncertain bi-level programming approach to formulate the incentive contract design model for the WRR intermodal transportation under uncertainty. In the following discussion, we explicitly elaborate each part of the formulation, including the notations, decision variables, assumptions, objective functions and constraints.

### Model Formulation and Notations

To formally characterize the problem of interest, we introduce the following notations to be used hereafter:

Decision variables:

$\left(w_{0i},w_{1i}\right)$: the parameters of incentive contract *i*, where $w_{0i}$ represents the fixed payment and $w_{1i}$ captures the incentive coefficient;

$e_{i}$: the carrier *i*’ effort level for reducing the travel time of transportation stage *i*, such as assigning more drivers and upgrading means of transport.

Parameters:

$T_{i}\left(e_{i},\xi_{i}\right)$: the travel time of transportation stage *i* (non-negative uncertain variable);

$C_{i}\left(e_{i}\right)$: the effort cost incurred at transportation stage *i*;

$W_{i}\left(T_{i}\right)$: the carrier *i*’ incentive contract offered by MTO;

*T*: the delivery time which is the total travel time, i.e., $T=\sum_{i=1}^{3}T_{i}$;

R(T): the MTO’s revenue;

$\pi_{i}^{0}$: the carrier *i*’ reservation utility.

Assumptions:

Throughout this paper, the following assumptions are made in our model formulation.

(i)We assume that $T_{i}\left(e_{i},\xi_{i}\right)=t_{0i}-t_{1i}e_{i}+\xi_{i}$, i=1,2,3, where $t_{0i}>0$ denotes the scheduled travel time of transportation stage *i* for the all contracted cargoes, $t_{1i}>0$ measures the marginal impact of the carrier *i*’ effort on shortening the announced travel time and $\xi_{i}$ characterizes the uncertainty.(ii)We assume that $\xi_{i}=L\left(-a_{i},b_{i}\right)$, i=1,2,3, are mutually independent linear uncertain variables with parameters left-width $a_{i}>0$ and right-width $b_{i}>0$.(iii)We assume that the MTO designs the time-based incentive contracts to induce the carriers to exert adequate efforts to complete their respective transportation stages, i.e., $W_{i}\left(T_{i}\right)=w_{0i}-w_{1i}T_{i}$, i=1,2,3.(iv)We suppose that $C_{i}\left(e_{i}\right)=\frac{1}{2}\lambda_{i}e_{i}^{2}$, i=1,2,3, where $\lambda_{i}>0$ represents cost coefficient. That is, when the carrier inputs effort level in the respective transportation stage, he incurs a cost to shorten the travel time.(v)We suppose that the MTO’s revenue depends linearly on the delivery time, i.e., $R\left(T\right)=r_{0}-r_{1}T$, where $r_{0}$ indicates the maximum revenue for the entire carriage and $r_{1}$ means the loss per unit time.

In Assumption i, we define a linear function for modeling the relationship between the travel time and the effort level. We believe that reality is more complicated. However, this linear function can make analytical solutions accessible, thus enabling us to derive the closed form expression of the optimal time-based incentive contract mechanism. Due to the lack of historical data, we use a linear uncertain variable for computational purposes to reflect the uncertainty in Assumption ii. Note that the linear uncertain variable is optional since some common uncertain variables such as zigzag uncertain variable can be applied. Since the causes of travel time uncertainties are unrelated (i.e., the carriers are engaging in different transportation stages), it is reasonable to assume that the uncertain components are independent. In Assumption iii, this time-based incentive contract has been used in practice to induce the subcontractors to increase their effort levels and complete their tasks earlier as proposed by Weitzman [41] and has been used by many researchers and practitioners, see Holmstrom and Milgrom [42]. In Assumption iv, a quadratic cost function is made not only for expositional convenience but also in accordance with the practical fact, which has been used in Yang et al. [39] and Tang et al. [43]. Assumption v is reasonable, because the WRR intermodal transport is a time-limit system for the delivery. That is, the reduction of delivery time by WRR intermodal transportation will lead to a reduction in financing costs, simply because the interest payment period will be made shorter after the delivery time is shorter.

Objective functions:

According to the parties’ different philosophies of modeling uncertainty, the expected utility and the mean-entropy utility functions (see Appendix B for details) are adopted to maximize the MTO’s and carriers’ profits, respectively.

*Expected utility*: The main idea of the expected utility is to optimize the expected value of the MTO’s profit, which can be expressed as
(1)$$\begin{array}{c} \Pi =E\left[R\left(\sum_{i=1}^{3}T_{i}\left(e_{i},\xi_{i}\right)\right)-\sum_{i=1}^{3}W_{i}\left(T_{i}\left(e_{i},\xi_{i}\right)\right)\right], \end{array}$$
which is equal to her expected revenue minus her payment for three carriers.

*Mean-entropy utility*: The essential idea of the mean-entropy utility is to optimize the linear combination of expected value and entropy of the carrier’s profit, which can be written as
(2)$$\pi_{i}=E\left[W_{i}\left(T_{i}\left(e_{i},\xi_{i}\right)\right)-C_{i}\left(e_{i}\right)\right]-\rho_{i}H\left[W_{i}\left(T_{i}\left(e_{i},\xi_{i}\right)\right)-C_{i}\left(e_{i}\right)\right],i=1,2,3,$$
where $\rho_{i}>0$ is the carrier *i*’s coefficient of risk aversion.

Constraints:

The individual rationality (IR) constraints, which guarantee the participation from the carriers, can be expressed as
(3)$$E\left[W_{i}\left(T_{i}\left(e_{i},\xi_{i}\right)\right)-C_{i}\left(e_{i}\right)\right]-\rho_{i}H\left[W_{i}\left(T_{i}\left(e_{i},\xi_{i}\right)\right)-C_{i}\left(e_{i}\right)\right]\geq \pi_{i}^{0},i=1,2,3.$$

The incentive compatibility (IC) constraints, which induce the carriers to improve effort levels, can be expressed as
(4)$$e_{i}\in \arg\max_{e_{i}^{\prime }\geq 0}\left(E\left[W_{i}\left(T_{i}\left(e_{i},\xi_{i}\right)\right)-C_{i}\left(e_{i}\right)\right]-\rho_{i}H\left[W_{i}\left(T_{i}\left(e_{i},\xi_{i}\right)\right)-C_{i}\left(e_{i}\right)\right]\right),i=1,2,3.$$

Using the above objectives and constraints, the incentive contract design problem for the WRR intermodal transportation can be stated as the following uncertain bi-level programming model: (5)$$\begin{array}{c} \left\{\begin{array}{c} \max_{\left(w_{0},w_{1}\right)}\Pi =E\left[R\left(\sum_{i=1}^{3}T_{i}\left(e_{i}^{*},\xi_{i}\right)\right)-\sum_{i=1}^{3}W_{i}\left(T_{i}\left(e_{i}^{*},\xi_{i}\right)\right)\right]\\ \mathrm{subject}\mathrm{to}:\\ \;\;E\left[W_{i}\left(T_{i}\left(e_{i}^{*},\xi_{i}\right)\right)-C_{i}\left(e_{i}^{*}\right)\right]-\rho_{i}H\left[W_{i}\left(T_{i}\left(e_{i}^{*},\xi_{i}\right)\right)-C_{i}\left(e_{i}^{*}\right)\right]\geq \pi_{i}^{0},i=1,2,3\;\\ \;\;w_{0i}\geq 0,i=1,2,3\;\\ \;\;w_{1i}\geq 0,i=1,2,3,\;\\ \;\;\left(e_{1}^{*},e_{2}^{*},e_{3}^{*}\right)\mathrm{solves}\mathrm{the}\mathrm{following}\mathrm{problem}P_{i},i=1,2,3\\ \;\;P_{i}:\;\left\{\begin{array}{c} \max_{e_{i}}\pi_{i}=E\left[W_{i}\left(T_{i}\left(e_{i},\xi_{i}\right)\right)-C_{i}\left(e_{i}\right)\right]-\rho_{i}H\left[W_{i}\left(T_{i}\left(e_{i},\xi_{i}\right)\right)-C_{i}\left(e_{i}\right)\right]\\ \mathrm{subject}\mathrm{to}:\\ \;\;e_{i}\geq 0. \end{array}\right. \end{array}\right. \end{array}$$

Our formulation is based on the model developed by Wang et al. [38]. Several novelties made by this paper relative to Wang et al. [38] are emphasized here. Firstly, in view of the studied problems, this paper studies a WRR intermodal transportation problem which is very different from the problem proposed in Wang et al. [38] that considered the project management problem. Secondly, in terms of proposed models, we give a mean-entropy utility function to characterize the philosophy of modeling uncertainty, but the entropy decision criterion was not used in Wang et al. [38]. Thirdly, based on the bi-level model, we study the impacts of model parameters on the information value of the effort (see Section 3.3), which was not discussed by Wang et al. [38]. By considering these three features, we take the first initiative to analyze the cooperation between the MTO and the carriers from a Stackelberg game of view, which contributes a new perspective to intermodal transportation theory and entropy theory.

In this paper, we analyze two cases to derive insights into which contract structures are appropriate in different information structures and investigate the impact of asymmetric information. For easy reference, we label the Cases S and A that will be studied in this paper. The first Case S represents the symmetric information scenario in which the carrier’s effort level is public information. The second Case A represents the asymmetric information scenario in which the carrier’s effort level is private information. In the following section, we will study each of these scenarios separately.

## 3. Optimal Time-Based Incentive Contracts Design

### 3.1. Symmetric Information Case

To explore the influence of asymmetric information, as a benchmark we first derive the optimal time-based incentive contracts when the MTO can directly contract on the carriers’ effort levels *e*. Thus, the IC constraints (4) in Model (Equation 5) are no longer required. Under symmetric information, the MTO specifies the effort levels *e* to optimize her expected profit for the carriers by writing a contract $\left(w_{0},w_{1}\right)$ that solves the following mathematical programming model: (6)$$\begin{array}{c} \left\{\begin{array}{c} \max_{\left(w_{0},w_{1},e\right)}\Pi^{S}=E\left[R\left(\sum_{i=1}^{3}T_{i}\left(e_{i},\xi_{i}\right)\right)-\sum_{i=1}^{3}W_{i}\left(T_{i}\left(e_{i},\xi_{i}\right)\right)\right]\\ \mathrm{subject}\mathrm{to}:\\ \;\;E\left[W_{i}\left(T_{i}\left(e_{i},\xi_{i}\right)\right)-C_{i}\left(e_{i}\right)\right]-\rho_{i}H\left[W_{i}\left(T_{i}\left(e_{i},\xi_{i}\right)\right)-C_{i}\left(e_{i}\right)\right]\geq \pi_{i}^{0},i=1,2,3\;\\ \;\;w_{0i}\geq 0,i=1,2,3\;\\ \;\;w_{1i}\geq 0,i=1,2,3.\; \end{array}\right. \end{array}$$

By computational methods for the expected value and Lemma A2, the MTO’s expected profit is defined in Theorem 1.

**Theorem** **1.***Assume that the travel time of three transportation stages are mutually independent linear uncertain variables. Under Case S, the objective function of Model (Equation 6) can be written as*
$$\Pi^{S}=E\left[\sum_{i=1}^{3}R\left(T_{i}\left(e_{i},\xi_{i}\right)\right)\right]-E\left[\sum_{i=1}^{3}W_{i}\left(T_{i}\left(e_{i},\xi_{i}\right)\right)\right]-2r_{0}.$$

**Proof.** See Appendix C. □

According to Theorem 1 and Lemma A3, we first decompose the Model (Equation 6) into three uncertain sub-models, and then derive the deterministic equivalent mathematical model in the following corollaries.

**Corollary** **1.***By Theorem 1, Model (Equation 6) can be decomposed into the following three sub-models:*
(7)$$\begin{array}{c} \left\{\begin{array}{c} \max_{\left(w_{0i},w_{1i},e_{i}\right)}\Pi_{i}^{S}=E\left[R\left(T_{i}\left(e_{i},\xi_{i}\right)\right)\right]-E\left[W_{i}\left(T_{i}\left(e_{i},\xi_{i}\right)\right)\right]\\ subjectto:\\ \;\;E\left[W_{i}\left(T_{i}\left(e_{i},\xi_{i}\right)\right)-C_{i}\left(e_{i}\right)\right]-\rho_{i}H\left[W_{i}\left(T_{i}\left(e_{i},\xi_{i}\right)\right)-C_{i}\left(e_{i}\right)\right]\geq \pi_{i}^{0}\;\\ \;\;w_{0i}\geq 0\\ \;\;w_{1i}\geq 0, \end{array}\right. \end{array}$$
*where i=1,2,3 and $\Pi^{S}=\Pi_{1}^{S}+\Pi_{2}^{S}+\Pi_{3}^{S}-2r_{0}$.*

**Corollary** **2.***By Lemma A3, the sub-model (Equation 7) can be transformed to the following deterministic mathematical programming:*
(8)$$\begin{array}{c} \left\{\begin{array}{c} \max_{\left(w_{0i},w_{1i},e_{i}\right)}\Pi_{i}^{S}=r_{0}-w_{0i}-\frac{1}{2}\left(r_{1}-w_{1i}\right)\left(2\left(t_{0i}-t_{1i}e_{i}\right)-a_{i}+b_{i}\right)\\ subjectto:\\ \;\;w_{0i}-\frac{1}{2}w_{1i}\left(2\left(t_{0i}-t_{1i}e_{i}\right)-a_{i}+b_{i}\right)-\frac{1}{2}\lambda_{i}e_{i}^{2}-\frac{1}{2}\rho_{i}w_{1i}\left(b_{i}+a_{i}\right)\geq \pi_{i}^{0},\;\\ \;\;w_{0i}\geq 0\\ \;\;w_{1i}\geq 0,\; \end{array}\right. \end{array}$$
*where i=1,2,3.*

Recognizing that the IR constraint of sub-model (Equation 8) will bind at the optimum and the carrier receives his reservation profit under Case S, we can design an efficient algorithm denoted as Algorithm 1 to solve sub-model (Equation 8). This procedure is described below:
**Algorithm 1**1:Shows that the IR constraint of the sub-model (Equation 8) is binding.2:Finds the optimal effort level by substituting the expected payment into the objective function of the sub-model (Equation 8) with the first-order condition.

Theorem 2 shows that the optimal time-based incentive contracts for the WRR intermodal transportation defined by Algorithm 1 under symmetric information case.

**Theorem** **2.***Under Case S, the MTO’s optimal time-based incentive contracts are given by*
$$\begin{array}{c} w_{1i}^{*}=0, \end{array}$$
$$\begin{array}{c} w_{0i}^{*}=\pi_{i}^{0}, \end{array}$$
*for i=1,2,3.*

**Proof.** See Appendix C. □

Theorem 2 gives closed form of the optimal time-based incentive contract mechanisms $\left(w_{0}^{*},w_{1}^{*}\right)$ under Case S. It is worth noting that the optimal incentive term $w_{1}^{*}$, regardless of risk aversion levels, is equal to 0 and the MTO sets the same values of $w_{1}^{*}$ to the carriers in this case. This can be explained as follows: when MTO can directly contract on the carriers’ effort levels, she does not need to motivate the carriers. In other words, the symmetric information case can be considered as a centralized one. Unlike the incentive coefficient $w_{1}^{*}$, the base payment $w_{0}^{*}$ in the optimal incentive contract depends on their reservation utilities. The intuition behind this is that the fixed payment is set to make sure that the carriers would accept the optimal time-based incentive contracts. According to Theorem 2, the corresponding effort level for carrier *i* under Case S is given as $e_{i}^{*}=\frac{r_{1}t_{1i}}{\lambda_{i}}$, i=1,2,3. Because the MTO can always request the effort level to be set at her optimum, there is no hidden effort level problem.

In short, the results under this case serve as a reference for the deriving results of the alternative cases studied. In this way, we can see why the asymmetric information is crucial to the optimal time-based incentive contract and further investigate the impact of asymmetric information on the optimal time-based incentive contracts for WRR intermodal transportation.

### 3.2. Asymmetric Information Case

Under asymmetric information, the carrier’s effort level for respective transportation stage is his private information. In this case, the carrier sets his effort level so as to maximize his own mean-entropy utility because it cannot be imposed as part of the contract terms. Under Case A, the MTO has to ensure that the contracts are incentive compatible in Model (Equation 5).

Similar to our analysis of the symmetric information in Section 3.1, we may encounter the difficulty of calculating the expected value and entropy value in the upper-level and lower-level programming problems in Model (Equation 5), respectively. To overcome these difficulties, we transform the bi-level programming model into its deterministic equivalent one.

**Theorem** **3.***Assume that the travel times of three transportation stages are mutually independent linear uncertain variables. Under Case A, the objective function of the upper-level problem of Model (Equation 5) can be written as*
$$\Pi^{A}=E\left[\sum_{i=1}^{3}R\left(T_{i}\left(e_{i}^{*},\xi_{i}\right)\right)\right]-E\left[\sum_{i=1}^{3}W_{i}\left(T_{i}\left(e_{i}^{*},\xi_{i}\right)\right)\right]-2r_{0}.$$

**Proof.** See Appendix C. □

On the basis of Theorem 3 and Lemma A3, the deterministic equivalent mathematical model is derived in the following corollaries.

**Corollary** **3.***According to Theorem 3, Model (Equation 5) can be decomposed into the following three sub-models:*
(9)$$\begin{array}{c} \left\{\begin{array}{c} \max_{\left(w_{0i},w_{1i}\right)}\Pi_{i}^{A}=E\left[R\left(T_{i}\left(e_{i}^{*},\xi_{i}\right)\right)\right]-E\left[W_{i}\left(T_{i}\left(e_{i}^{*},\xi_{i}\right)\right)\right]\\ subjectto:\\ \;\;E\left[W_{i}\left(T_{i}\left(e_{i}^{*},\xi_{i}\right)\right)-C_{i}\left(e_{i}^{*}\right)\right]-\rho_{i}H\left[W_{i}\left(T_{i}\left(e_{i}^{*},\xi_{i}\right)\right)-C_{i}\left(e_{i}^{*}\right)\right]\geq \pi_{i}^{0}\;\\ \;\;w_{0i}\geq 0\;\\ \;\;w_{1i}\geq 0,\;\\ \;\;e_{i}^{*}solvesproblemsthefollowingproblemP_{i},\\ \;\;P_{i}:\;\left\{\begin{array}{c} \max_{e_{i}}\pi_{i}^{A}=E\left[W_{i}\left(T_{i}\left(e_{i},\xi_{i}\right)\right)-C_{i}\left(e_{i}\right)\right]-\rho_{i}H\left[W_{i}\left(T_{i}\left(e_{i},\xi_{i}\right)\right)-C_{i}\left(e_{i}\right)\right]\\ subjectto:\\ \;\;e_{i}\geq 0. \end{array}\right. \end{array}\right. \end{array}$$
*where i=1,2,3 and $\Pi^{A}=\Pi_{1}^{A}+\Pi_{2}^{A}+\Pi_{3}^{A}-2r_{0}$.*

**Corollary** **4.***According to Lemma A3, the uncertain bi-level programming sub-model (Equation 9) an be transformed to the following deterministic mathematical programming:*
(10)$$\begin{array}{c} \left\{\begin{array}{c} \max_{\left(w_{0i},w_{1i}\right)}\Pi_{i}^{A}=r_{0}-w_{0i}-\frac{1}{2}\left(r_{1}-w_{1i}\right)\left(2\left(t_{0i}-t_{1i}e_{i}^{*}\right)-a_{i}+b_{i}\right)\\ subjectto:\\ \;\;w_{0i}-\frac{1}{2}w_{1i}\left(2\left(t_{0i}-t_{1i}e_{i}^{*}\right)-a_{i}+b_{i}\right)-\frac{1}{2}\lambda_{i}{\left(e_{i}^{*}\right)}^{2}-\frac{1}{2}\rho_{i}w_{1i}\left(b_{i}+a_{i}\right)\geq \pi_{i}^{0}\;\\ \;\;w_{0i}\geq 0\;\\ \;\;w_{1i}\geq 0,\;\\ \;\;e_{i}^{*}solvesproblemsthefollowingproblemP_{i},\\ \;\;P_{i}:\;\left\{\begin{array}{c} \max_{e_{i}}\pi_{i}^{A}=w_{0i}-\frac{1}{2}w_{1i}\left(2\left(t_{0i}-t_{1i}e_{i}\right)-a_{i}+b_{i}\right)-\frac{1}{2}\lambda_{i}e_{i}^{2}-\frac{1}{2}\rho_{i}w_{1i}\left(b_{i}+a_{i}\right)\\ subjectto:\\ \;\;e_{i}\geq 0. \end{array}\right. \end{array}\right. \end{array}$$

Taking advantage of the structural characteristics of the deterministic model (Equation 10), we suggest an iterative Algorithm 2 to solve each deterministic sub-problem. The solution process can be divided into two steps, which is described as follows:
**Algorithm 2**1:Determine the optimal solution of the lower-level programming problem by using the first-order condition and substitute it into the IR constraint in the upper-level programming problem.2:Show that the IR constraint is binding and substitute the expected payment into the objective function of the upper-level programming problem of which the optimal solution is easily obtained by using the first-order condition.

By Algorithm 2, we can obtain the following theorem characterizing the optimal time-based incentive contracts offered by the MTO under asymmetric information case.

**Theorem** **4.***Under Case A, the MTO’s optimal time-based incentive contracts are given by*
$$\begin{array}{c} w_{1i}^{**}=r_{1}-\frac{\lambda_{i}\rho_{i}\left(a_{i}+b_{i}\right)}{2t_{1i}^{2}}, \end{array}$$
$$\begin{array}{c} w_{0i}^{**}=\frac{1}{2}w_{1i}^{**}\left(2t_{0i}+\left(\rho_{i}-1\right)a_{i}+\left(\rho_{i}+1\right)b_{i}\right)-\frac{1}{2}{\left(w_{1i}^{**}\right)}^{2}\frac{t_{1i}^{2}}{\lambda_{i}}+\pi_{i}^{0}, \end{array}$$
*for i=1,2,3.*

**Proof.** See Appendix C. □

Theorem 4 provides us with closed form expression of the optimal time-based incentive contract mechanisms $\left(w_{0}^{**},w_{1}^{**}\right)$ under Case A. This theorem interprets that the carrier *i*’s risk aversion level $\rho_{i}$ has an impact on the optimal time-based incentive contract mechanism $\left(w_{0i}^{**},w_{1i}^{**}\right)$. The managerial insight of Theorem 4 is that the MTO should design the incentive contract for the carrier *i* based on his risk aversion level $\rho_{i}$ in the decision making process. By Theorem 4, the corresponding effort level for carrier *i* under Case A is given as $e_{i}^{**}=\frac{r_{1}t_{1i}}{\lambda_{i}}-\frac{\rho_{i}\left(a_{i}+b_{i}\right)}{2t_{1i}}$, i=1,2,3. In can be seen that the optimal effort level $e_{i}^{**}$ decreases in $\rho_{i}$. That is, the MTO has to induce the conservative carriers to improve effort level to accelerate their respective transportation stages.

### 3.3. Effect of Information Asymmetry

In this subsection, we draw a comparison between the optimal time-based incentive contract offered by the MTO under symmetric and asymmetric information cases to understand the effect of information asymmetry. It is intuitive that information asymmetry will introduce distortions in the optimal decisions for the MTO and the carriers.

**Corollary** **5.**By comparing the optimal time-based incentive contract under symmetric information with that under asymmetric information, we can obtain $w_{1i}^{*}<w_{1i}^{**}$, for i=1,2,3.

Corollary 5 shows that the optimal incentive coefficient under asymmetric information is greater than its optimal value in the benchmark-setting of symmetric information. The optimal incentive term is distorted upward by $\frac{\lambda_{i}\rho_{i}\left(a_{i}+b_{i}\right)}{2t_{1i}^{2}}$ under Case A. This is an expected result because the carrier must be provided with information rent, which distorts the incentive term upward.

**Corollary** **6.**The carrier i’s optimal effort level associated with the symmetric information case and asymmetric information case satisfy $e_{i}^{*}>e_{i}^{**}$, for i=1,2,3.

The intuition behind Corollary 6 is as follows. The optimal time-based incentive contract under the symmetric information case results in higher effort levels and hence shortens the travel time of transportation compared to the optimal time-based incentive contract under asymmetric information case.

## 4. Simulation Analysis

To explore the characteristics of the model, we present a simulation example to study and analyze the proposed models. With the reported results, we believe that these findings have significant implications for practice and indicate the need for time-based incentive contracts. In this section, we consider a WRR intermodal transportation instance, in which the water, rail and road transportation stages are performed by carriers in sequence. In particular, an MTO plans, organizes, and funds the entire intermodal transportation chain and subcontracts each transportation stage to the respective specialized carriers.

From the above-mentioned discussion, it can be inferred that we can make the obtained solution more realistic by addressing the uncertainties. Specifically, the travel time of each transportation stage depending on the carrier’s effort level is unknown and characterized as an uncertain variable via the experts’ estimations. Without loss of generality, using a linear uncertain variable $T_{i}=L\left(t_{0i}-t_{1i}e_{i}-a_{i},t_{0i}-t_{1i}e_{i}+b_{i}\right)$ to denote the travel time uncertainty with uncertainty distribution
$$\begin{array}{cc} \Phi (x)\;\;\;\; & =\left\{\begin{array}{cc} 0, & \;x<t_{0i}-t_{1i}e_{i}-a_{i}\\ \frac{x-t_{0i}+t_{1i}e_{i}+a_{i}}{a_{i}+b_{i}}, & \;t_{0i}-t_{1i}e_{i}-a_{i}\leq x\leq t_{0i}-t_{1i}e_{i}+b_{i}\\ 1, & \;x>t_{0i}-t_{1i}e_{i}+b_{i}, \end{array}\right. \end{array}$$
for i=1,2,3. In this simulation example, we set $t_{01}=48$ h, $t_{11}=1$ (h/effort), $t_{02}=36$ h, $t_{12}=2$ (h/effort), $t_{03}=24$ h, $t_{13}=3$ (h/effort). For the sake of convenience, we set $a_{1}=a_{2}=a_{3}=2$ h and $b_{1}=b_{2}=b_{3}=1$ h, which are used for computational purposes to reflect the uncertainty in asymmetrical forms.

Facing travel time uncertainty, we consider differing perceptions of risk for the MTO and carriers. More specifically, the risk-neutral MTO tends to maximize the expected profit based on the expected utility and the risk-averse carriers maximize their certainty equivalent values which take the mean-entropy form. For the sake of convenience, we set $\rho_{1}=\rho_{2}=\rho_{3}=\rho$, which is the parameter characterizing carriers’ risk aversion levels. As mentioned in the previous section, the carrier *i* must bear the effort cost $C_{i}=\frac{1}{2}\lambda_{i}e_{i}^{2}$, i=1,2,3. Due to the difference in the transportation condition, we assume that $\lambda_{1}=1$ (dollar/effort${}^{2}$), $\lambda_{2}=3$ (dollar/effort${}^{2}$) and $\lambda_{3}=5$ (dollar/effort${}^{2}$). Taking into carriers’ outside opportunity consideration, we let $\pi_{1}^{0}=100$ (dollar), $\pi_{2}^{0}=300$ (dollar) and $\pi_{3}^{0}=500$ (dollar) be the corresponding reservation utilities.

As a matter of fact, the MTO can earn a higher profit if the entire transportation process is completed faster. Based on this, considering that revenue decreases in the delivery time, we assume that $R=3000-10(T_{1}+T_{2}+T_{3})$. On the other hand, the MTO often cannot observe how much effort the carriers use to reduce the travel time of their transportation stages. As a result, the MTO needs to offer the time-based incentive contracts to the carriers for ensuring that the cargo will move to their destination as fast as possible.

### 4.1. Computational Results

By Theorems 2 and 4, we can obtain the closed form expressions for optimal time-based incentive contract and effort levels for the carriers under Cases S and A, which are shown in Table 1. Table 1 illustrates that the optimal time-based incentive contract mechanisms $\left(w_{01}^{**},w_{11}^{**}\right)$, $\left(w_{02}^{**},w_{12}^{**}\right)$ and $\left(w_{03}^{**},w_{13}^{**}\right)$ are determined by the carriers’ risk aversion levels ρ. Therefore, these obtained results can provide useful guidance to the MTO for adjusting incentives coefficients based on the carriers’ risk aversion levels.

Next, we investigate the impact of the parameter ρ alteration on MTO’s expected profit under Cases S and A. Figure 2 shows that the MTO’s expected profit $\Pi^{A}$ relies on the risk aversion level ρ. More specifically, red and blue lines represent the MTO’s expected profits $\Pi^{S}$ and $\Pi^{A}$, respectively. In particular, we examine the information value, which is defined as the difference between the MTO’s expected profits with and without contracting on the carriers’ effort levels, i.e., $IV=\Pi^{S}-\Pi^{A}$. Figure 2 demonstrates that the MTO can always benefit when she can contract on the carriers’ effort levels. Therefore, the results suggest that from the MTO’s perspective, it is beneficial to have better information about the carriers’ effort levels.

### 4.2. Discussion

Based on the computational results, we give several managerial insights and interpretations:In can be seen from Table 1 that the incentive coefficients decrease with the risk aversion level. This is an expected result because if the carriers are more conservative, the MTO should motivate the carriers to exert higher effort levels.It can be observed from Figure 2 that as the risk aversion level ρ increases, the MTO’s expected profit $\Pi^{A}$ decreases while $\Pi^{S}$ remains the same. The rationale is that the MTO will lose more profits when the carriers become more conservative.As shown in Figure 2, we can find that as the risk aversion level ρ increases, the value of information IV is increasing. That is, the more conservative the carriers are, the greater the value of information will be. The intuitive explanation for this result is that acquiring information of the effort becomes important for the MTO if the carriers become more conservative.

## 5. Conclusions and Future Research

In this paper, we addressed the uncertain incentive contract design problem with an MTO and three carriers working in sequence in a WRR intermodal transportation setting. For modeling travel time uncertainty, we formulated an uncertain bi-level programming model for the incentive contract design problem under the expectation and the entropy decision criteria based on the uncertainty theory. Taking advantage of the structural characteristics, we first divided the original model into three sub-models and then used an iterative algorithm to derive closed-form expressions for the optimal time-based incentive contracts. To broaden the scope of application of this model, we implemented a simulation example in a WRR intermodal transportation setting. The computational results show that the MTO is more willing to acquire the carrier’s effort level information under the uncertain travel time environment by using the numerical analysis.

Finally, we conclude by providing some directions for future research. In this paper, we assumed that the travel times of the transportation stages are mutually independent. A further research path could be to study the case where the travel times are correlated. Another alternative setting is one where the carrier may wish to utilize her resources more effectively by varying her effort level over time, which might extend our static model to the dynamic environment. Last but not least, the other heuristic algorithms for increasing the performance of achievements could also be considered in future research.

## Figures and Tables

**Figure 1 entropy-21-00161-f001:**
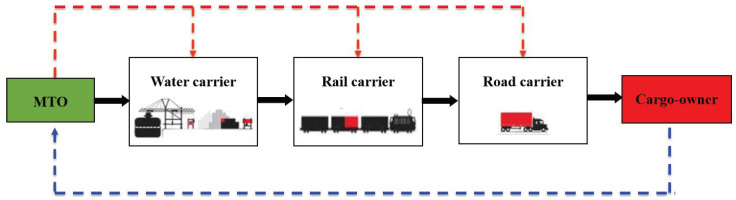
The sequence of the events in water-rail-road (WRR) intermodal transportation.

**Figure 2 entropy-21-00161-f002:**
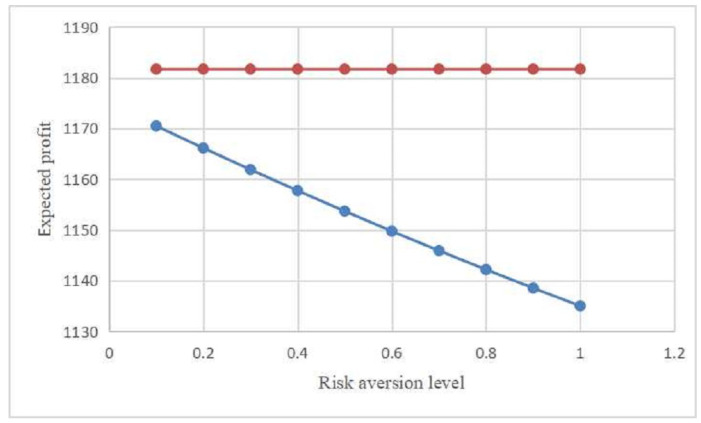
Impact of risk aversion level on the multimodel transport operator’s (MTO’s) expected profit.

**Table 1 entropy-21-00161-t001:** Closed form expressions for optimal time-based incentive contract mechanism and effort level.

Scenario	Carrier	$w_{0}$	$w_{1}$	*e*
	Water	100	0	10
Symmetric information	Rail	300	0	$\frac{20}{3}$
	Road	500	0	6
	Water	$\frac{1}{2}\left(10-\frac{3}{2}\rho \right)\left(95+3\rho \right)-\frac{1}{2}{\left(10-\frac{3}{2}\rho \right)}^{2}+100$	$10-\frac{3}{2}\rho$	$10-\frac{3}{2}\rho$
Asymmetric information	Rail	$\frac{1}{2}\left(10-\frac{9}{4}\right)\left(71+3\rho \right)-\frac{2}{3}{\left(10-\frac{9}{4}\right)}^{2}+300$	$10-\frac{9}{8}\rho$	$\frac{20}{3}-\frac{3}{4}\rho$
	Road	$\frac{1}{2}\left(10-\frac{5}{2}\rho \right)\left(47+3\rho \right)-\frac{9}{10}{\left(10-\frac{5}{2}\rho \right)}^{2}+500$	$10-\frac{5}{2}\rho$	$6-\frac{1}{2}\rho$

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
