# Peer review of "Incentive Contract Design for the Water-Rail-Road Intermodal Transportation with Travel Time Uncertainty: A Stackelberg Game Approach"

_entropy, 2019, doi:10.3390/e21020161_

Round 1

Reviewer 1 Report

The overall issue addressed in this paper is interesting. However, I have the following comments (mainly on the model and the theory):

Page 1: check "In respond to decentralized decision making process"

Page 2: empirical examples are needed to support the statements "As a matter of fact, there exists information asymmetry between the MTO and the carriers. That is, the work rate is not only the carrier’s private information and unknown to the MTO, but it also at a cost to the carrier, and the carrier may not provide work rate that the MTO desires. This may lead to conflict between the carriers and the MTO and delay the delivery time."

Page 3: I am not convinced by the following statement: "uncertain travel time based upon expert’s subjective knowledge, experience and professional feelings that is again a valid assumption which holds in many real-life problems." In my view, travel time is more likely repeatable and why would subjective perception is better than objective historical data in such cases? Even the assumption is valid, why are the well-established fuzzy logic and fuzzy programming methods not used or not suitable? Why would uncertainty theory be more suitable than fuzzy programming methods?

Page 3: the research method is based on uncertain theory, which I am not familiar with. However, it appears that the majority of the supporting references on uncertain theory were not published in the mainstream mathematics and OR/MS journals. It may require a mathematician to validate the underlying theory.

Pages 4-5: the research context is not sufficiently explained and justified. It seems that the MTO is handling the container. If this is the case, there are several concerns about the model. Firstly, the travel time (by vessel or by train) determined by work rate is questionable, because vessel schedule and train schedule are fixed and announced well in advance. Secondly, containers may stay at ports/terminals for a significant number of days. The travel times of by road and by train could be much smaller than the time at ports. However, the time at ports is not considered in the model.

Page 5: the key concept “work rate” is not explicitly defined.

Page 5: the assumption of the linear function of travel time to work rate is questionable. For example, a truck can carry one TEU or two TEUs. That means, the travel time is the same for one TEU container or two TEU containers. A train and a vessel can carry many containers in one trip.

Page 5: because the key assumptions are not convincing, the model is questionable.

Page 11: there is no explanation where are the data coming from? It is unclear what is the unit of travel times, e.g. t01, t11, t02, t12, t13?

Page 11: there is no justification for the values of many parameters, e.g. a1, a2, a3, b1, b2, etc.

Pages 10-12: it is unclear whether the experiments are realistic.

Reviewer 2 Report

in the attached file

Reviewer 3 Report

The paper is aimed to explore new methodologies for design Intermodal Transport contracts for WRR intermodal transportation from the perspective of the Multimodal Transport Operator.

The paper is well-wrtitten and well-structured, the gaps related with the previous literature are clearly identified and the novel features of the study are clearly stated.

I retain the paper suitable for the publication, and suggest only certain minimal mistakes, as in the following.

Page 1: in the title there is a double word “INCENTIVE”

Page 2, row 34: FROM instead of FORM

Page 11, row 343: “THAT THE AS THE RISK”, revise

Page 5: figure 1 is not ery useful, consider to delete it

Page 12: a section including a discussion about the obtained results should be introduced after the concluding section

Round 2

Reviewer 1 Report

Re-review comments:

Comment 2: the response is not satisfactory. The newly added sentence does not provide empirical examples support the "information asymmetry between the MTO and the carriers".

Comment 3: the response is not satisfactory. First, are you assuming that your research applies to the first time contract between MTO and carrier? However, I think the solution to a game model is the result of interactive behaviours of the players. Is the modelling method appropriate for the problem?

Second, the claim "But some surveys showed that the subjective uncertainty cannot be modelled by fuzzy numbers (variables)" is vague and not supported. More importantly, the authors have not explained why fuzzy numbers are not suitable for the problem under consideration. What does self-duality mean and why would self-duality be essential in your research problem?

Comment 5: the response is not satisfactory. The decision variable e_i is changed from "work rate" to "effort level". The change appears to be arbitrary and does not reflect the nature of the original problem. More importantly, it raises new questions. For example, the new definition of e_i (i.e. carrier's effort level) is rather intangible and difficult to measure. How could you quantify the effort and how could you establish the relationship between travel time and the effort level. Why would they have linear relationship? 

You have included port time as part of T_i(e_i, xi_i). However, port time is normally depending on port/terminal operators and/or customs operations. How could carrier's effort determine port time?  

Comment 6: the change of the definition "work rate" is questionable. It appears that the utility has been changed to be per TEU. However, is it realistic that MTO and carrier will sign contract on individual TEU basis even the shipment has multiple TEUs?

Comment 7: The authors did not address the comment why there is a linear relationship. 

Comment 8: this comment still stands.

Comment 9: the authors did not answer the first question.

Comment 10: no satisfactory justification is given

Comment 11: not addressed.

Author Response

Thank you once again for your time in reviewing this manuscript and your suggestions for improving and revising the paper. Please see the attachment for more details.

Round 3

Reviewer 1 Report

The paper has been improved.

There are a lot of assumptions to simplify the problem.

The authors should list all these assumptions explicitly, and explain their rationale and discuss implications somewhere in the paper.

Author Response

(The authors gave the same response as above.)
